# Hi-fi functional priors by learning activations

**Marcin Sendera** [1,2,3] *   **Amin Sorkhei**   **Tomasz Kuśmierczyk** [1] *
[1] Jagiellonian University   [2] Mila   [3] Université de Montréal

## Abstract

Function-space priors in Bayesian Neural Networks (BNNs) provide a more intuitive approach to embedding beliefs directly into the model's output, thereby enhancing regularization, uncertainty quantification, and risk-aware decision-making. However, imposing function-space priors on BNNs is challenging. We address this task through optimization techniques that explore how trainable activations can accommodate higher-complexity priors and match intricate target function distributions. We investigate flexible activation models, including Pade functions and piecewise linear functions, and discuss the learning challenges related to identifiability, loss construction, and symmetries. Our empirical findings indicate that even BNNs with a single wide hidden layer when equipped with flexible trainable activation, can effectively achieve desired function-space priors.

## 1   Introduction

Models trained in a function-space rather than in the space of weights and biases (parameters space) exhibit flatter minima, better generalization, and improved robustness to overfitting (Qiu et al., 2024). Better properties achieved by focusing directly on the output space are particularly advantageous in scenarios where the relationship between parameters and function behavior is not straightforward, especially for Bayesian Neural Networks (BNNs). Moreover, function-space priors offer a direct method of specifying beliefs about the functions being modeled by BNNs, rather than just the parameters, leading to intuitive and often more meaningful representation of prior knowledge (Tran et al., 2022).

Finding accurate posteriors for deep and complex models such as BNNs is notoriously challenging due to their high-dimensional parameter spaces and complex likelihood surfaces. On the other hand, for single-hidden-layer wide BNNs, it has been recently shown that posterior sampling via Markov Chain Monte Carlo (MCMC) can be performed efficiently (Hron et al., 2022). Moreover, research has demonstrated that the exact posterior of a wide BNN weakly converges to the posterior corresponding to the Gaussian Process (GP) that matches the BNN's prior (Hron et al., 2020). This allows BNNs to inherit GP-like properties while preserving their advantages. For example, BNNs scale better to large datasets as they can also leverage deep learning methods, reducing computational burden compared to GPs. The relation between NNs and BNNs has been a topic of significant research interest, which we briefly discuss in Section A.

A classic result by Neal (1996); Williams (1996), later extended to deep NNs by Lee et al. (2017); Matthews et al. (2018), shows that infinitely wide layers in NNs behave *a priori* like GPs by identifying the kernel of a GP that matches covariance of a (B)NN. The past studies were conducted for better understanding of NNs. We are interested in a similar setting, but our objective is the opposite: we aim to implement the function-space priors (in particular, GP-like behavior) in BNNs, by matching *both* their priors on parameters and activations. GP prior specification generally offers greater interpretability compared to the prior applied to the weights of a BNN. This is because the kernel

---

* denotes equal contribution

Workshop on Bayesian Decision-making and Uncertainty, 38th Conference on Neural Information Processing Systems (NeurIPS 2024).

clearly governs key characteristics of the prior functions, such as shape, variability, and smoothness. We provide an extended reasoning on BNNs mitigating GPs behaviour in Appendix B.

Finding BNNs that exhibit the behavior of GPs is a notoriously difficult problem, with analytical solutions existing for only a few GP kernels. For example, Meronen et al. (2020) found a solution (a BNN's activation function) for the popular Matern kernel. To address the challenge of imposing function-space *a priori behavior* to BNNs, Flam-Shepherd et al. (2017), Flam-Shepherd et al. (2018), and especially Tran et al. (2022), attempted to find both parameters and priors on parameters by using gradient-based optimization. Their approach, which requires deep networks equipped with complex priors, presents an additional undisclosed challenge in learning posteriors. Instead, we demonstrate that by learning also activations, we can achieve faithful function-space priors using just a shallow, wide BNN. In this work, we establish a practical and straightforward alternative to finding closed-form solutions (activations) to impose function-space priors on BNNs.

Results in the paper are supplemented in the Appendix, which covers Related Work (Section A), a discussion on the relationship between BNNs and GPs (Section B), challenges in optimization (e.g., identifiability; Section C), and details of the Experimental Setting followed by Additional Results.

## 2   Finding weights and activations to match function-space priors

The covariance between two inputs $x$ and $x'$ of a (single-hidden layer wide) BNN is

$$\text{cov}(f^l(x), f^l(x')) = \kappa_f^l(x, x') = \sigma_b^{l\,2} + \sigma_w^{l\,2} \cdot \mathbb{E}_{f_j^{l-1}}[\phi(f_j^{l-1}(x))\phi(f_j^{l-1}(x'))] \tag{1}$$

where $l$ reads here as *last* (or output) and $l-1$ as *input* layer. The BNN matches, a priori, a GP with an appropriate kernel $\kappa$ realised by the covariance cov. In this work, we focus on zero-centered i.i.d. priors for weights and biases, e.g., $\mathbb{E}[w] = 0$ and $\mathbb{E}[b] = 0$. To ensure that the asymptotic variance neither vanishes nor explodes, marginal variances are assumed to be inversely proportional to layer width, i.e., $\mathbb{V}[w_{ij}^l] = \frac{\sigma_w^{l\,2}}{H_l}$. For the full discussion, please see Section B.

We consider the inverse problem of imposing behavior akin to a $\text{GP}(0, \kappa)$ on a BNN. The task is to identify hyperparameters in Eq. 1, i.e., priors on $w$, $b$ (where $f^{l-1}$ is also expressed via $w^{l-1}$ and $b^{l-1}$) and activation $\phi$, to align them with the desired GP, such that on an input subspace (=*index set*) $\mathcal{X}$, the covariance would match the kernel. Ideally, $f^l \sim \text{GP}(0, \kappa)_{[\mathcal{X}]}$, i.e., $p_{nn}(f^l(X)) = p_{gp}(f_i^l(X))$ for any $X \subset \mathcal{X}$. However, in practice we strive for the approximate similarity: $p_{nn}(f_i^l(X)) \approx p_{gp}(f_i^l(X))$. The matching is performed on an input subspace denoted here by the functional-space measurement set $X$ (=$N$ input points of dimensionality $F$, which can generally be defined in multiple ways (Sun et al., 2019)) and is posed as an optimization task: $p_{nn}^* = \text{argmin}_{p_{nn}} \frac{1}{S} \sum_{X \sim p_X} D(p_{nn}(f_i^l(X)), p_{gp}(f_i^l(X)))$, where $D$ is an *arbitrary* (but in our case differentiable) divergence measure, and we average over $S$ samples from the input space. While $p_{gp}$ can be both sampled and evaluated (for fixed inputs $X$, the GP reduces to a Gaussian distribution), $p_{nn}$ is known only implicitly, which means that we can sample from it but its density cannot be evaluated directly.

We proceed by reparameterizing the prior distributions on model weights and biases. In particular, we use the basic zero-centered factorized Gaussians with learned variances, e.g., $p(w|\sigma_w^2)$, $p(b|\sigma_b^2)$. However, we assume additionally, parametric and differentiable activations $\phi(\cdot|\eta)$. Thus, $p_{nn}$ is fully specified by $\lambda = \{\sigma, \eta\}$, leading to the final optimization objective:

$$\lambda^* = \text{argmin}_\lambda \frac{1}{S} \sum_{X \sim p_X} D(p_{nn}(f^l(X)|\lambda), p_{gp}(f^l(X))) \tag{2}$$

which we address through gradient-based optimization with respect to $\lambda$. This formulation involves deciding on the loss function $D$ and the model for $\phi$.

The problem of learning activations for NNs involves designing functions that enable networks to capture complex and non-linear relationships effectively. In principle, any function $\phi : \mathcal{R} \to \mathcal{R}$ can serve as an activation, but the choice significantly impacts the network's expressiveness and training dynamics. We explore several models for $\phi$, including Rational (Pade) (Molina et al., 2019), Piece-wise linear (PWL)[1], and activations realized by a NN with a single narrow hidden layer and

---

[1]https://pypi.org/project/torchpwl/

ReLU/SiLU own activations. These functions are efficient to compute and introduce desirable non-linear properties, striking a balance between efficiency and the capacity to model intricate patterns.

The loss $D$ measures the divergence between two distributions over functions. Due to the implicit nature of $p_{nn}$, it must be specified between sets of samples $\{f_i\}$ (we used 512 samples from each distribution) and approximated using Monte Carlo. We discovered that the standard losses, including KL-divergence, fail for this task. For example, in KL, estimating the empirical entropy term presents a challenge. Consequently, we followed Tran et al. (2022) and relied on the Wasserstein distance:

$$D = \left( \inf_{\gamma \in \Gamma(p_{nn}, p_{gp})} \int_{\mathcal{F} \times \mathcal{F}} d(f, f')^p \gamma(f, f') df df' \right)^{1/p}$$
$$= \sup_{|\psi|_L \leq 1} \mathbb{E}_{p_{nn}}[\psi(f)] - \mathbb{E}_{p_{gp}}[\psi(f)] \tag{3}$$

Unlike Tran et al. (2022), we used the 2-Wasserstein metric, which for Multivariate Gaussians (applicable in the case of GPs and wide BNNs, at least approximately) has a closed-form solution (Mallasto and Feragen, 2017):

$$D = ||\mu_1 - \mu_2||_2^2 + \text{Tr}\left( \Sigma_1 + \Sigma_2 - 2\sqrt{\sqrt{\Sigma_1} \Sigma_2 \sqrt{\Sigma_1}} \right),$$

where $\mu_{1/2}$, $\Sigma_{1/2}$ are respectively expectations and covariance matrices estimated for $p_{nn}$ and $p_{gp}$ from samples $\{f^l(X)\}$ obtained at finite input sets $X$. Not only we avoid the internal optimisation due to $\sup_{|\psi|}$, but additionally $D$ can be efficiently computed based on results by Buzuti and Thomaz (2023). For experiments, we report numerical values of $D$ normalized by number of elements in $X$. If the assumptions cannot be fulfilled, we can always revert back to the nested optimization for $|\psi|_L$ in Eq. 3.

## 3 Learning activations enables more accurate function-space priors.

To map GP function-space priors to a BNN, we can: train its priors on parameters, train both priors and activation, or learn just the activation function. We empirically show that learning also activations provides better solutions to the inverse problem of imposing function-space priors on BNNs. Figure 1 illustrates the results of an extensive study where we compare the quality of the GP prior fit for various models of parameters' priors and activations. The target was $GP(0, \text{Matern}(\nu = 5/2, l = 1))$. For each configuration, we conducted several training iterations and measured the final (converged) loss value multiple times. Generally, learning activations alone is not sufficient for good fits, but when combined with learning priors, it significantly improves the solutions.

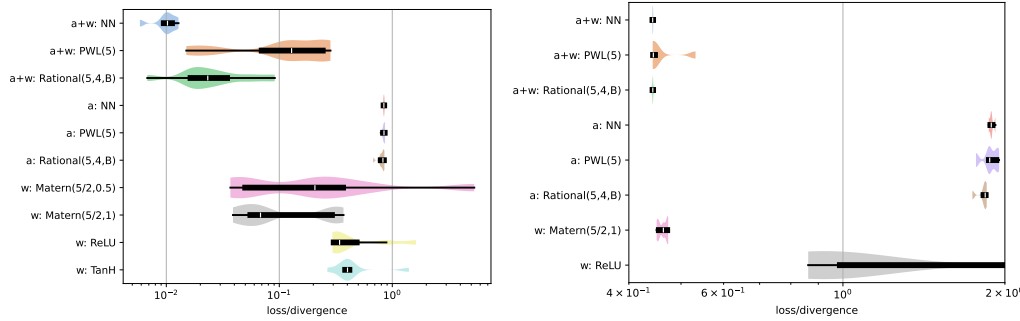

Figure 1: Quality of matching BNNs to the prior of a GP with a Matern kernel ($\nu = 5/2$, *length scale*=1.0) for 1D inputs (left) and 16D inputs (right). We evaluate models with trained parameter priors (denoted by *w*), activations (denoted by *a*), and both (denoted by *a+w*). Each label specifies whether a fixed activation (e.g., ReLU) or a specific activation model (e.g., Rational) was employed. Gaussian parameter priors were used by default. If not trained, we set variances to 1., and for the hidden layer, we normalized the variance by its width. The label *w: Matern* refers to a BNN with the closed-form (fixed) activation as derived by Meronen et al. (2020).

# 4 Do expressive function-space priors require networks to be deep?

Tran et al. (2022) takes a practical approach to learning function-space priors. Instead of considering BNNs with wide layers, they *postulate* that a deep network as a whole can model a functional prior. They then tune its parameters to reflect this functional prior. This involves modeling complex priors on all weights considered jointly, for example, using Normalizing Flows (Rezende and Mohamed, 2015) to ensure sufficient fidelity. This approach reveals a challenge in finding posteriors for deep networks. While MCMC (Chen et al., 2014; Del Moral et al., 2006) algorithms are computationally expensive, approximate posteriors (Hoffman et al., 2013; Ritter et al., 2018) do not possess sufficient expressiveness to adapt to such complex priors. We demonstrate that a BNN with a single wide hidden layer, basic Gaussian priors, and learned activation can match or even outperform the results of Tran et al. (2022), both in terms of priors and posteriors quality. For the description of the experimental setting, see Appendix D. We present the results in Fig. 2 and in Tab. 1 (in Appendix).

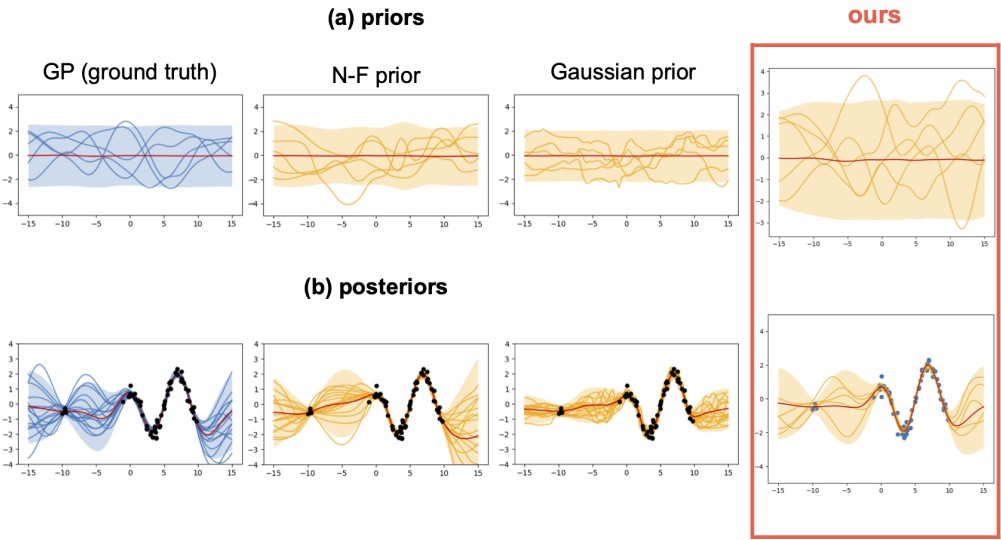

Figure 2: Prior ((**a**)) and posterior ((**b**)) predictive distributions for a BNN with trained parameters priors and activations (ours; 4th column), and for Tran et al. (2022) approach with different prior realizations (Gaussian - 3rd column and Normalizing Flow - 2nd). The first column illustrates the ground truth (GP). Numerical results complementing the figures we provide in Tab. 1 in Appendix.

# 5 Can learned activations match performance of the closed-form ones?

Complexity of deriving suitable neural activations makes matching BNNs to GPs a formidable challenge. We instead propose a gradient-based learning method that adapts both activations and parameters priors. This approach departs from the traditional methods that rely on fixed, analytically derived activations, offering instead a flexible alternative that empirically can match or even surpass the performance of the more conventional approaches.

The empirical evaluation we performed for a problem of 2D data classification following Meronen et al. (2020), with a GP, where the authors derived analytical activation to match the Matern kernel exactly. For our method, we used a BNN with Gaussian priors with trained variances, where the activation function was modeled by a NN with a single hidden layer consisting of 5 neurons using SiLU activation. Posterior distributions were generally obtained using a HMC sampler. For the comprehensive description of the setting, please see Appendix D.

Tab. 2 and Fig. 3 demonstrate that our method enhances the match between the posteriors of a BNN and the desired target GP. Note that Meronen et al. (2020) originally used MC Dropout to obtain the posteriors, achieving much worse results than those obtained with HMC. For fairness in comparison, we tested both MC Dropout and HMC.In terms of performance, our model not only captures class probabilities accurately but also adeptly handles the total variance in class predictions and the epistemic uncertainty component, which are crucial for robust decision-making under uncertainty.

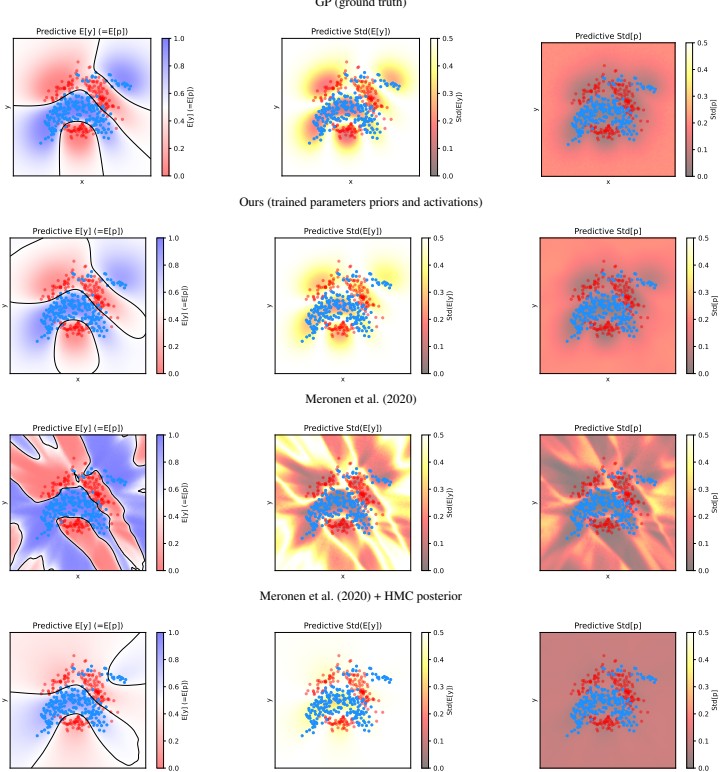

Figure 3: Posterior predictive distributions for a BNN with trained parameters priors and activations (ours; 2nd row), and for a BNN with analytically derived activations with posteriors determined using MC Dropout (3rd row) and HMC (4th row). The first column illustrates class probabilities, the second column shows the total variance in class predictions, and the last column depicts the epistemic uncertainty component of the total uncertainty. Numerical results complementing the figures we provide in Tab. 2 in Appendix.

## 6 Conclusion

In this paper, we addressed the problem of transferring functional priors for wide Bayesian Neural Networks to replicate desired a priori properties of Gaussian Processes. Previous approaches typically focused on learning distributions over weights and biases, often requiring deep BNNs for sufficient flexibility. We proposed an alternative approach by also learning activations, providing greater adaptability even for shallow models and eradicating the need for task-specific architectural designs. To the best of our knowledge, we are the first to explore learning activations in this context.

## Acknowledgements

This research is part of the project No. **2022/45/P/ST6/02969** co-funded by the National Science Centre and the European Union Framework Programme for Research and Innovation Horizon 2020 under the Marie Skłodowska-Curie grant agreement No. 945339. For the purpose of Open Access, the author has applied a CC-BY public copyright licence to any Author Accepted Manuscript (AAM) version arising from this submission.

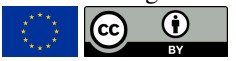

This research was in part funded by National Science Centre, Poland, **2022/45/N/ST6/03374**. We gratefully acknowledge Polish high-performance computing infrastructure PLGrid (HPC Center: ACK Cyfronet AGH) for providing computer facilities and support within computational grant no. **PLG/2023/016302**.

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

# A  Related work

The relation between NNs/BNNs has been a topic of significant research interest. For example, Meronen et al. (2021) explored periodic activation functions in BNNs to connect network weight priors with translation-invariant GP priors. Pearce et al. (2020) derived BNN architectures to mirror GP kernel combinations, showcasing how BNNs can produce periodic kernels. Furthermore, Karaletsos and Bui (2020) introduced a hierarchical model using GP for weights to encode correlated weight structures and input-dependent weight priors, aimed at regularizing the function space. Matsubara et al. (2021) proposed using ridgelet transforms to approximate GP function-space distributions with BNN weight-space distributions, providing a non-asymptotic analysis with finite sample-size error bounds. Finally, Tsuchida et al. (2019) extended the convergence of NN function distributions to GPs under broader conditions, including partially exchangeable priors. For a more detailed discussion of related topics see for example, Sections 2.3 and 4.2 in Fortuin (2021). Here, we included just a brief review of the selected works.

# B  On BNNs imitating GPs

Behavior akin to GPs has been *observed* or *postulated* within neural networks. In particular, let's consider a Multilayer Perceptron:

$$f_i^0(x) = \sum_j^I w_{ij}^0 x_j + b_i^0, \;\; i = 1 \dots H_0; \;\; f_i^l(x) = \sum_j^{H_l} w_{ij}^l \phi(f_j^{l-1}(x)) + b_i^l, \;\; i = 1 \dots H_l \quad (4)$$

where $x$ denotes $I$-dimensional input, and $f^L$ denotes $O$-dimensional output. In BNNs, $z_{ij}(x) = w_{ij}^l \phi(f_j^{l-1}(x))$ is a r.v. and assuming common prior distributions for $w$ and $b$, $f_i^l(x)$ becomes a sum of i.i.d r.v.-s. Then, from Central Llimit Theorem follows $p(f_i^l(x)) \xrightarrow{H^l \to \infty} \mathcal{N}(\mu(x), \sigma^2(x))$. For two different inputs $x, x'$ we can write $p(f^l(x), f^l(x')) \xrightarrow{H^l \to \infty} \mathcal{N}(\mu_f^l[x, x'], \kappa_f^l[x, x'])$ where we use $[\cdot]$ to denote that we limit the mean and covariance potentially specified on a bigger space to these two inputs. In particular, by Kolmogorov extension theorem, the prior distribution of functions induced in the $l$-th layer, weakly converges to a GP as $p(f^l) \xrightarrow{H^l \to \infty} \text{GP}(\mu_f^l, \kappa_f^l)$.

We focus on zero-centered i.i.d. priors for weights and biases, e.g., $\mathbb{E}[w_{ij}^l] = 0$ and $\mathbb{E}[b^l] = 0$. Furthermore, to ensure that asymptotic variance neither vanishes nor explodes, marginal variances are assumed to be inversely proportional to layer width, i.e., $\mathbb{V}[w_{ij}^l] = \frac{\sigma_w^{l\,2}}{H_l}$. Then, following from the definition, the covariance for two inputs $x$ and $x'$ can be obtained as (we repeat here Eq. 1)

$$\text{cov}(f^l(x), f^l(x')) = \kappa_f^l(x, x') = \sigma_b^{l\,2} + \sigma_w^{l\,2} \cdot \mathbb{E}_{f_j^{l-1}}[\phi(f_j^{l-1}(x))\phi(f_j^{l-1}(x'))]$$

The expectation is taken over realisations of the r.v. $f_i^{l-1}$. We write $\mathbb{E}_{f_j^{l-1}}$ meaning $\mathbb{E}_{p(f_j^{l-1}(x)), p(f_j^{l-1}(x'))}$ to signify r.v.-s which are integrated out.

MLPs consisting of several (wide) hidden layers were considered by Lee et al. (2017) and Matthews et al. (2018). Such the network can be considered a compound GP, where outputs of each layer are modeled by a GP. Then, $f_i^{l-1} \sim \text{GP}(0, \kappa^{l-1})$ and subsequent ($l$-th) layer pre-activations $(f_j^l, f_k^l)^T \sim \mathcal{N}(0, \Sigma^l \cdot \delta_{jk})$ where $\Sigma^l = \kappa_f^l(x, x') = \sigma_b^{l\,2} + \sigma_w^{l\,2} \cdot \mathbb{E}_{f_i^{l-1} \sim \text{GP}(0, \kappa^{l-1})}[\phi(f_i^{l-1}(x))\phi(f_i^{l-1}(x'))]$. For an even more basic network with only one wide hidden layer, as considered by Neal (1996); Williams (1996), the behavior converges to that of a single individual GP, inheriting its capacity for *arbitrary function approximation* (Rasmussen and Williams, 2005). The approximation capability will depend on the chosen activation function, the nature and scale of the prior distributions on the weights and biases, and the width of the layer. For this case, the r.v.-s $f_i^0$ are expressed with $w^0$ and $b^0$ and the expectation in Eq.(1) is taken over these variables as $\mathbb{E}_{f_i^0}[\cdot] = \mathbb{E}_{w^0 \sim p_w^0, b_i^0 \sim p_b^0}[\cdot]$.

# C  Selected challenges in optimizing BNNs with learnable activations

Models with a larger number of hyperparameters and a higher degree of freedom allow for achieving more complex distributions and potentially better solutions to Eq.(2). On the other hand, overparam-

eterization and complication of shape of the optimization manifold, may create identifiability problems or increase the risk of getting stuck in a local optimum for the steepest-gradient optimization.

To alleviate this, the optimization task in Eq.2 can be slightly simplified by appropriately fixing parameters in Eq.(1). In particular, bias $b_i^l$ can be removed (set to 0) and variance of weights between layer $l-1$ and $l$ can be set to $\mathbb{V}[w_{ij}^l] = \frac{1}{H_l}$ effectively simplifying Eq.(1) to

$$\text{cov}(f^l(x), f^l(x')) = \kappa^l(x, x') = \mathbb{E}_{f_i^{l-1}}[\phi(f_i^{l-1}(x))\phi(f_i^{l-1}(x'))] \tag{5}$$

We cannot fix the weights as we propose to do for biases as this would mean that $f_i^l(x) = f_j^l(x)$ effectively reducing such the layer to a single output.

On the other hand, the problem of finding BNNs behaving like GPs (e.g. inverting covariance equation) in unidenfiable. There exist multiple solutions assuring similar quality of the final match. In particular, activations $\phi$ solving Eq.(2) are not unique, regardless if $p_{nn}$ is given by Eq.(1) or Eq.(5) (below). For example:

- The solutions are symmetric w.r.t activity values (y-axis), i.e., for $\phi'(f) = -\phi(f)$, values of covariance given by Eq.(1) or Eq.(5) are not changed, simply because $(-1)^2 = 1$.
- For $p(f_i^{l-1}(x))$ symmetric around 0 (for example, Gaussians), activations with flipped arguments $\phi'(f) = \phi(-f)$ result in the same covariances.
- Scaling activations $\phi'(f) = \alpha\phi(f)$ leads to the same covariances as scaling variances of the output weights as $(\sigma_w^l)' = \alpha \cdot \sigma_w^l$

Given a sufficiently flexible model (like a neural network itself), one can learn to approximate any target activation function to an arbitrary degree of accuracy on a compact domain. This is in line with the universal approximation theorem. However, in practice there are multiple limitations and challenges. A model may require an impractically large number of parameters to approximate certain complex functions to a desired level of accuracy. More complex models may be harder to fit and may require more training data. Some functions might require high numerical precision to be approximated effectively, and even if a model can fit a target activation function on a compact set, it might not generalize well outside the training domain. Overall, one needs to consider factors like the gradient behavior (for backpropagation), computational efficiency, and numerical stability. These factors can limit the practicality of using certain models of activations.

## D  Experimental setting

**1-dimensional regression - comparison with Tran et al. (2022)**

In order to show the abilities of our approach on a standard regression task and for a fair comparison with another method optimizing Wasserstein distance, we follow the exact experimental setting presented in Tran et al. (2022). We used a GP with RBF kernel (*length scale*=0.6, *amplitude*=1.0 and *noise variance*=0.1) as the ground truth.

For a baseline, we used a method from Tran et al. (2022) consisting of a BNN (3 layers with 50 neurons each) with TanH activation function. We consider all three priors realizations presented in that work: simple Gaussian prior, hierarchical prior and prior given by a Normalizing Flow. On the other hand, our model configuration consists of a BNN with Gaussian priors centered in **0** with trained variances, where the activation function was modeled by a NN with a single hidden layer consisting of 5 neurons using SiLU activation. Posterior distributions were obtained using a HMC sampler.

We find that our approach allows for achieving the same or better results than Tran et al. (2022) baseline both by visual and numerical comparisons. In general, we obtain better results in terms of distributional metrics as presented in Tab. 1 and getting better (visually) posterior distributions (see Fig.2) without utilizing any specific activation function or computationally heavy prior realization (like, e.g., Normalizing Flow).

**Classification - comparison with Meronen et al. (2020)**

For the comparison against Meronen et al. (2020), we followed their experimental setting and used a GP with Matern kernel ($\nu = 5/2$, *length scale*=1) as the ground truth. For a baseline we utilized

the method by Meronen et al. (2020), where the authors derived analytical activation to match the Matern kernel. Our model configuration consists of a BNN with Gaussian priors with trained variances, where the activation function was modeled by a NN with a single hidden layer consisting of 5 neurons using SiLU activation. Posterior distributions were generally obtained using a HMC sampler, except in the case of (Meronen et al., 2020), where the original implementation employed MC Dropout to approximate the posterior. However, for completeness and fairness in comparison, we also generated results using a HMC-derived posterior for the model by Meronen et al. (2020). Additional tests (see Tab. 2) were conducted on BNNs with fixed parameters priors (*Default*=Gaussian priors with variance of 1, and *Normal*=Gaussian priors normalized by the hidden layer width) and with activation functions including ReLU and TanH.

# E    Numerical evaluation

In this section, we present the comprehensive numerical evaluation for the experiments from the main paper. The results for regression experiments are presented in Tab. 1, whereas the results for classification are presented in Tab. 2.

Table 1: Comparison of the similarity of trained (function-space) priors and posteriors in a 1D regression task between Tran et al. (2022) and our method. Whereas Tran et al. (2022) considered three different prior (on parameters) models (including Gaussian, Hierarchical, and Normalizing Flow) for deep BNNs, our implementation consists of a single-hidden-layer BNN with standard Gaussian priors with learned variances and trained activation. The methods were compared using a set of distributional metrics against the ground truth provided by a GP (following the code by Tran et al. (2022)). We compared both prior and posterior predictive distributions of functions over a range of $\mathcal{X}$ covering regions with and without data (to account also for overconfidence far from data). The lower, the better.

| Setting | 1-Wasserstein | 2-Wasserstein | Linear-MMD | Poly-MMD $\times 10^3$ | RBF-MMD | Mean-MSE | Mean-L2 | Mean-L1 | Median-MSE | Median-L2 | Median-L1 |
|---|---|---|---|---|---|---|---|---|---|---|---|
| ***Priors:*** | | | | | | | | | | | |
| Gaussian prior | 7.49 | 7.55 | -3.48 | 1.225 | 0.033 | 0.003 | 0.055 | 0.048 | 0.007 | 0.084 | 0.070 |
| Hierarchical prior | 8.14 | 8.37 | 0.22 | 1.421 | 0.027 | 0.004 | 0.064 | 0.059 | 0.008 | 0.090 | 0.074 |
| Normalizing Flow prior | 7.76 | 8.16 | -2.36 | 0.388 | 0.021 | 0.005 | 0.071 | 0.057 | 0.006 | 0.076 | 0.057 |
| **ours** | 7.02 | 7.21 | -8.68 | 0.088 | 0.008 | 0.004 | 0.062 | 0.053 | 0.007 | 0.086 | 0.074 |
| ***Posteriors:*** | | | | | | | | | | | |
| Gaussian prior | 6.59 | 6.78 | 23.45 | 5.272 | 0.26 | 0.259 | 0.509 | 0.332 | 0.294 | 0.542 | 0.354 |
| Hierarchical prior | 5.61 | 5.84 | 15.12 | 3.1 | 0.153 | 0.112 | 0.335 | 0.216 | 0.122 | 0.349 | 0.219 |
| Normalizing Flow prior | 6.81 | 7.00 | 34.19 | 7.044 | 0.263 | 0.275 | 0.524 | 0.383 | 0.368 | 0.607 | 0.423 |
| **ours** | 10.76 | 10.91 | 78.66 | 8.133 | 0.739 | 0.781 | 0.883 | 0.703 | 0.794 | 0.891 | 0.713 |

Table 2: Similarity of the posterior predictive distributions of BNNs with a single wide hidden layer to the posterior of a GP, considered as ground truth. Calculations were performed for a test grid that includes regions both with and without training data to account for overconfidence far from data. The posteriors were obtained using the HMC sampler in all cases, except for (Meronen et al., 2020), where the original code was used (however, results for this model with a HMC-derived posterior are also included below). The lower, the better. For our method, we present results using weights and priors pretrained for 1D inputs, as well as those trained on functional priors for 1D/2D inputs matched to the task on $\mathcal{X}$ (see Fig. 3).

| Setting | 1-Wasserstein | 2-Wasserstein | Linear-MMD | Mean-L1 | Mean-L2 | Mean-MSE | Median-L1 | Median-L2 | Median-MSE | Poly-MMD$\times 10^3$ | RBF-MMD |
|---|---|---|---|---|---|---|---|---|---|---|---|
| Default with ReLU | 39 | 39 | 667 | 0.23 | 0.26 | 0.067 | 0.3 | 0.33 | 0.11 | 4518 | 0.16 |
| Default with TanH | 38 | 38 | 595 | 0.22 | 0.24 | 0.058 | 0.3 | 0.32 | 0.11 | 4181 | 0.15 |
| Normal with ReLU | 31 | 32 | 603 | 0.21 | 0.25 | 0.061 | 0.22 | 0.26 | 0.069 | 3093 | 0.47 |
| Normal with TanH | 25 | 25 | 314 | 0.14 | 0.18 | 0.031 | 0.15 | 0.18 | 0.034 | 1721 | 0.55 |
| (Meronen et al., 2020) | 36 | 36 | 777 | 0.24 | 0.28 | 0.078 | 0.28 | 0.32 | 0.1 | 7311 | 0.24 |
| Normal + HMC | 21 | 21 | 72 | 0.07 | 0.09 | 0.007 | 0.071 | 0.094 | 0.009 | 435 | 0.24 |
| Default + HMC | 42 | 42 | 284 | 0.14 | 0.16 | 0.026 | 0.32 | 0.34 | 0.12 | 1867 | 0.11 |
| **Ours: pretrained 1D** | 25 | 25 | 73 | 0.06 | 0.09 | 0.008 | 0.076 | 0.1 | 0.011 | 771 | 0.07 |
| **Ours: trained 1D** | 23 | 23 | 6 | 0.03 | 0.03 | 0.001 | 0.029 | 0.035 | 0.001 | 45 | 0.03 |
| **Ours: trained 2D** | 23 | 23 | 13 | 0.03 | 0.03 | 0.001 | 0.028 | 0.035 | 0.001 | 74 | 0.02 |

