# OpenReview forum: "Hi-fi functional priors by learning activations"
_NeurIPS.cc/2024/Workshop/BDU — NeurIPS BDU Workshop 2024 Poster_

### Official Review · Reviewer_U687 · 2024-09-26
**Interesting ideas, acceptable presentation**

**Rating:** 7
**Confidence:** 3

**Review:**

The paper proposes a novel approach to learning activation functions by minimizing a loss function between the GP probability and the NN probability. Below are my major comments:

The paper lacks a discussion on how this approach compares to prior works that integrate domain knowledge, such as Dylan Sam et al.'s Bayesian Neural Networks with Domain Knowledge Priors (ArXiv:2402.13410).

Typically, researchers transition from GP to BNN when the input space becomes large. How does this approach scale with respect to the size of the input space?

---

### Official Review · Reviewer_kg5H · 2024-10-07
**Good paper.**

**Rating:** 6
**Confidence:** 3

**Review:**

Strengths:

1. The paper is well organized, and the presentation is good.

2. The theoretical contribution of this paper is sufficient, which is a bright spot.

Weaknesses:

1. If more experimental results in the appendix can be added to the main text, the experimental part will be more complete.

2. At the end of this paper, the summary of the full text seems to be missing.

3. In the relevant work, the investigation of previous studies is insufficient.

---

### Decision · Program_Chairs · 2024-10-09

Accept (Poster)